

# Physiological, biochemical and phytohormone responses of *Elymus nutans* to α-pinene-induced allelopathy

Mengci Chen[1], Youming Qiao[1], Xiaolong Quan[1], Huilan Shi[2] and Zhonghua Duan[1]

[1] State Key Laboratory of Plateau Ecology and Agriculture, Qinghai University, Xining, Qinghai, China
[2] College of Ecol-Environmental Engineering, Qinghai University, Xining, Qinghai, China

Corresponding author
Youming Qiao, ymqiao@aliyun.com

## ABSTRACT

The α-pinene is the main allelochemical of many weeds that inhibit the growth of *Elymus nutans*, an important forage and ecological restoration herbage. However, the response changes of α-pinene-induced allelopathy to *E. nutans* is still unclear. Here, we investigated the physiological, biochemical and phytohormone changes of *E. nutans* exposed to different α-pinene concentrations. The α-pinene-stress had no significant effect on height and fresh weight (FW) of seedlings. The water-soluble proteins, the soluble sugars and proline (Pro) strengthened seedlings immunity at 5 and 10 $\mu L\,L^{-1}$ α-pinene. Superoxide dismutase (SOD) and ascorbate peroxidase (APX) increased at 5 $\mu L\,L^{-1}$ α-pinene to resist stress. APX reduced the membrane lipid peroxidation quickly at 10 $\mu L\,L^{-1}$ α-pinene. The high-activity of peroxidase (POD), APX along with the high level of GSH contributed to the cellular redox equilibrium at 15 $\mu L\,L^{-1}$ α-pinene. The POD, glutathione reductase (GR) activity and glutathione (GSH) level remained stable at 20 $\mu L\,L^{-1}$ α-pinene. The changes in antioxidant enzymes and antioxidants indicated that *E. nutans* was effective in counteracting the harmful effects generated by hydrogen peroxide ($H_2O_2$). The α-pinene caused severe phytotoxic effects in *E. nutans* seedlings at 15 and 20 $\mu L\,L^{-1}$. Endogenous signal nitric oxide (NO) and cell membrane damage product Pro accumulated in leaves of *E. nutans* seedlings at 15 and 20 $\mu L\,L^{-1}$ α-pinene, while lipid peroxidation product malondialdehyde (MDA) accumulated. The chlorophylls (Chls), chlorophyll a (Chl a), chlorophyll b (Chl b) content decreased, and biomass of seedlings was severely inhibited at 20 $\mu L\,L^{-1}$ α-pinene. The α-pinene caused phytotoxic effects on *E. nutans* seedlings mainly through breaking the balance of the membrane system rather than with reactive oxygen species (ROS) productionat 15 and 20 $\mu L\,L^{-1}$ α-pinene. Additionally, phytohormone levels were altered by α-pinene-stress. Abscisic acid (ABA) and indole acetic acid (IAA) of *E. nutans* seedlings were sensitive to α-pinene. As for the degree of α-pinene stress, salicylic acid (SA) and jasmonic acid (JA) played an important role in resisting allelopathic effects at 15 $\mu L\,L^{-1}$ α-pinene. The ABA, Zeatin, SA, gibberellin 7 (GA7), JA and IAA levels increased at 20 $\mu L\,L^{-1}$ α-pinene. The α-pinene had a greatest impact on ABA and IAA levels. Collectively, our results suggest that *E. nutans* seedlings were effective in counteracting the harmful effects at 5 and 10 $\mu L\,L^{-1}$ α-pinene, and they were severely stressed at 15 and 20 $\mu L\,L^{-1}$ α-pinene. Our findings provided references for understanding the allelopathic mechanism about allelochemicals to plants.

## INTRODUCTION

Environmental change and grassland degradation in the Sanjiangyuan region of the Tibetan Plateau is one of the main issues that scientists have been concerned about for many years and has degraded significantly, manifested as grassland degradation and undersupply of pasture (*Wang, Long & Cao, 2006*; *Qin, 2014*). Allelopathy of many noxious and unpalatable plants is one of the important ecological mechanisms of grassland degradation. Many weeds produce allelochemicals that inhibit existing plants and are able to produce large numbers of seeds and compete vigorously for nutrients with forages (*Zhang et al., 1989*; *Guo et al., 2017*; *Shang et al., 2008*). Spread of weeds and allelopathic inhibition lead toward weeds further colonization and the ultimate degradation of grassland (*Shang et al., 2013*; *Ren, 2013*).

Drooping wildryegrass (*Elymus nutans*) is a native perennial grass and plays an important role in ecological restoration projects in the alpine meadow region of the Tibetan Plateau. It grows extensively in alpine and humid areas with an altitude of 2500~4000 m, and is distributed in Inner Mongolia, Qinghai, Tibet and Sichuan, China. Compared with other excellent germplasm resources that have been domesticated and selected for restoration of degraded grasslands, such as crymophila bluegrass (*Poa crymophila*) and Kentucky bluegrass (*Poa pratensis*), drooping wildryegrass has been used more widely and for longer periods of time (*Shang et al., 2018*). Meanwhile, drooping wildryegrass has high crude protein content and good palatability, which is suitable for supplementing pasture for livestock.

*Ajania tenuifolia* is one of the major weeds in seeded drooping wildryegrass grasslands, and is closely related to their degradation (*Ren, Shang & Long, 2014*). The $\alpha$-pinene is one of the main allelochemicals isolated from the volatile oil of *A. tenuifolia* (*Zhen et al., 1996*). As an important monoterpene substance (*Allenspach et al., 2020*), $\alpha$-pinene is the main secondary metabolite of the essential oil of many plants (*Adlard, 2010*). It is volatile and hydrophobic, with fresh rosin and woody aroma (*Pastore, Vespermann & Paulino, 2017*). The $\alpha$-pinene are released to the environment through volatilization (*Kamal, 2020*). At present, the research of allelopathy of weeds mainly uses the aqueous extracts, organic solvent extracts from plants (*Weston & Duke, 2003*; *Wang et al., 2021*). The extracts of plants containing $\alpha$-pinene had different degree of allelopathic inhibition on seeds germination and growth of other plants. The main essential oil in the leaves of *Vitex pseudo-negundo* at flowering stage were $\alpha$-pinene and $\alpha$-terpinyl acetate. The essential oil of vitex is associated with inhibitory effects on the seed germination and growth of *Lepidium sativum*, *Amaranthus retroflexus* and *Taraxacum officinale* (*Haghighi, Saharkhiz & Naddaf, 2019*). It is found that the main essential oil components of rosemary (*Rosmarinus officinalis*) at different phenological stages were $\alpha$-pinene. The inhibitory effect of essential oil was associated with seeds germination and growth of *Lactuca serriola* and *Rhaphanus sativus* at different concentrations (*Alipour & Saharkhiz, 2016*). In our previous

studies, it was also found that in the aqueous extracts of *Pedicularis kansuensis*, *Stellera chamaejasme*, *Elsholtzia densa* and *Morina chinensis*, the main weeds in the grasslands of the plateau region, had higher $\alpha$-pinene content. These plants together with *A. tenuifolia* release allelochemicals and inhibit growth of drooping wildryegrass in synergetic ways, and gradually caused degradation of alpine pastures (*Cheng et al., 2011*; *Liang et al., 2019*). Despite extracts of plants with a $\alpha$-pinene-base that have been reported to have allelopathic inhibition, little is known on the allelopathy of a single substance $\alpha$-pinene.

At present, there are limited reports about impact of allelopathy on phytohormone. No information is available on $\alpha$-pinene-induced allelopathy for drooping wildryegrass. In this study, we analyzed the allelopathic responses changes by investigating various indicators related to growth, photosynthesis, biochemical and phytohormone levels of drooping wildryegrass seedlings exposed to different $\alpha$-pinene concentrations in a hydroponic system. To our knowledge, this is the first time to study the allelopathic effects of $\alpha$-pinene in drooping wildryegrass seedlings from the physiological, biochemical and phytohormone profiles. Our findings also provided references for understanding the allelopathic mechanism of allelochemicals in plants.

## MATERIALS & METHODS

### Plant materials, growth conditions and treatments

Seeds of *E. nutans* were collected from Tongde Forage Seed Production Base of Qinghai Province (China; 35°15′N, 100°38′E) in September 2019. Seeds were surface sterilized with NaClO [ 0.5% (v/v)] for 15 min and washed 8 times with distilled water ($dH_2O$). The 1.5 gram of healthy seeds were germinated in sterilized a Petri dish with 4 ml distilled water. The germinated seeds were cultivated in a growth chamber with 12 h light and 12 h dark [photon density: 9000 Lux, diurnal temperature: $(25 \pm 2)$ / $(20 \pm 2)$ °C, relative humidity: 65–70%] using 1/2 Hoagland solution. The nutrient constituents of 1/2 Hoagland solution comprised $KNO_3$ (2.5 mmol/L), $Ca(NO_3)_2$ (2.5 mmol/L), $MgSO_4$ (one mmol/L), $NH_4H_2PO_4$ (0.5 mmol/L), NaFeEDTA (50 µmol/L), $H_3BO_3$ (7.5 µmol/L), $MnCl_2$ (1.25 µmol/L), $CuSO_4$ (0.5 µmol/L), $ZnSO_4$ (1 µmol/L). The nutrient solution was changed every day. After 14 days, healthy seedlings were treated by 0, 5, 10, 15 and 20 µL $L^{-1}$ $\alpha$-pinene (The concentration selected was based on the plant growth phenotype obtained from the results of previous pre-experiments) in the transparent closed tank. 0 µL $L^{-1}$ $\alpha$-pinene was the control treatment. The $\alpha$-pinene (>98% purity) were purchased from Macklin Company (China). The transparent closed tank of the same volume was inverted, so that various concentrations $\alpha$-pinene was added to the lid. A Petri dish with healthy seedlings was put in every transparent closed tank, only $\alpha$-pinene varied in concentration between 0 and 20 µL $L^{-1}$. The intention was for releasing $\alpha$-pinene to different concentration levels by volatilization into the transparent closed tank. The nutrient solution and $\alpha$-pinene were changed every day. Control and $\alpha$-pinene-treated seedlings continued to grow for 4 days under the above stated conditions. Leaves of drooping wildryegrass seedling were collected to determine the responses of the growth-related indicator related to physiological, biochemical and hormonal processes associated-indicators. Three independent replications of each treatment were used to determine each indicator.

## Shoot height, fresh weight and dry weight

The height, FW and dry weight (DW) of 15 shoot drooping wildryegrass seedlings were measured, weighed and soaked for each treatment, followed by an oven-drying at 80 °C for 48 h. Relative water content (RWC) of the shoot was calculated based on FW, DW and turgid weight (TW) (*Mostofa & Fujita, 2013*), formula for RWC (%) = $100 \times$ (FW−DW) / (TW−DW)

## Contents of water-soluble proteins, soluble sugars and photosynthetic pigments

The contents of water-soluble proteins and soluble sugars were determined in the fresh leaves of drooping wildryegrass by bicinchoninic acid (BCA) method (*Campion, Loughran & Walls, 2011*) and anthrone colorimetry (*Bai et al., 2013*). The leaves of drooping wildryegrass were extracted with 80% (V/V) acetone, and the absorbance of supernatant was recorded at 663 nm and 645 nm. The Chls, Chl a and Chl b contents were calculated according to the formula (*Arnon, 1949*).

## Malondialdehyde, hydrogen peroxide, proline, glutathione and nitric oxide contents

The contents of malondialdehyde (MDA) were determined in the fresh leaves of drooping wildryegrass by the thiobarbituric acid method, using MDA detection Kit (MDA-1-Y). $H_2O_2$ in drooping wildryegrass leaves were extracted by acetone and the contents were determined using the Kit $H_2O_2$-1-Y. The contents of Pro were determined by acidic ninhydrin method, using PRO detection Kit (PRO-1-Y). GSH contents were determined by 2-nitrobenzoic acid method, using GSH detection Kit (GSH-1-W). The contents of NO were determined by diazonium salt method,using NO-1-G kit. All the kits for measuring activities were purchased from Comin Biotechnology Co., Ltd., Suzhou, China (http://www.cominbio.com).

## Extraction and assays of enzymes

The activities of SOD, APX, POD, catalase (CAT), GR and nitrate reductase (NR) were determined in the fresh leaves of drooping wildryegrass seedlings under treatment. The six enzymes indexes were determined according to the manufacturer's protocol of assay kits SOD-1-W  for SOD activity; APX-1-W for APX activity; POD-1-Y for POD activity; CAT-1-Y for CAT activity; GR-1-W kit for GR activity; NR-1-W for NR activity. All the kits for activities were purchased from Comin Biotechnology Co., Ltd., Suzhou, China (http://www.cominbio.com).

## Phytohormone contents

The endogenous hormones in seedling leaves of drooping wildryegrass were measured with high-performance liquid chromatography tandem mass spectrometry (HPLC-MS/MS). The internal standards, including IAA, ABA, JA, SA, Zeatin, gibberellin 4 (GA4) and GA7, were purchased from Sigma-Aldrich (Burlington, MA, USA). Leaf samples were accurately weighed to 1 g and ground to powder in liquid nitrogen. Ten times the volume of acetonitrile and 8 μL internal standards was added to the powder, and then placed

**Table 1  Selected reaction monitoring conditions for protonated or deprotonated plant hormones([M +H]$^+$ or[M −H]$^-$).**

| Name | Electrode | Precursor ions (m/z) | Product ions (m/z) | Clustering voltage (V) | Collision energy (V) |
|---|---|---|---|---|---|
| ABA | − | 263.1 | 153.1[a]/204.2 | −60 | −14/−27 |
| GA4 | − | 331.1 | 243.2[a]/213.1 | −131 | −24/−39 |
| GA7 | − | 329.2 | 223.2[a]/241.1 | −89 | −38/−22 |
| IAA | + | 176.1 | 130.1[a]/102.9 | 65 | 12/42 |
| JA | − | 209.2 | 59.1[a] | −54 | −16 |
| SA | − | 137 | 92.9[a]/65 | −50 | −20/−39 |
| Zeatin | + | 220.4 | 148.1/136.0[a] | 92 | 22/16 |

**Notes.**
[a]Quantitative ion

at 4 °C a night. After centrifuge at 12,000 g for 5 min, the supernatant was extracted. Five times the volume of acetonitrile was added to the sediment. The supernatant was combined after extraction again, and added 35 mg C18 QuECherSmixed pack, mixed by shaking for 30 s. After centrifugation at 10,000 g for 5 min, the supernatant was extracted. The supernatant was dried with nitrogen, and dissolved in 400 μL methanol and passed through a 0.22 μm filter for HPLC-MS/MS. The samples were tested by HPLC (Aglient 1290, USA) coupled to a triple-stage quadrupole mass spectrometer (AB SCIEX-6500 Qtrap; SCIEX, Framingham, MA, USA) and used electrospray ionization (ESI) as the ion source for MRM detection mode scanning. The data of endogenous hormone was obtained using monitoring conditions for protonated or deprotonated plant hormones ([M+H]$^+$ or [M-H]$^-$) (Table 1).

## Statistical Analysis

The data were analyzed using IBM SPSS Statistics 21 software. Kruskal-Wallis Test was used to detect the differences. A $p$ value <0.05 was considered significant. The data were presented as mean ± standard error.

## RESULTS

### Effects of $\alpha$-pinene on plant growth, biomass, RWC, toxicity symptoms and photosynthetic pigment of drooping wildryegrass seedlings

The $\alpha$-pinene treatments had no significant influences on plant height and FW, but resulted in significant decrease in DW of seedling at 15 and 20 μL L$^{-1}$ $\alpha$-pinene ($\chi^2 = 11.567$, $df = 4$, $p = 0.021$; Table 2). The two $\alpha$-pinene concentrations also affected the water status of drooping wildryegrass seedlings. The RWC of leaves increased by 42.5, 41.1%, at 15, 20 μL L$^{-1}$ $\alpha$-pinene, respectively ($\chi^2 = 11.167$, $df = 4$, $p = 0.025$; Table 2). The leaves of drooping wildryegrass seedlings began to yellow 4 days after 20 μL L$^{-1}$ $\alpha$-pinene treatment (Fig. 1). Consistent with phenotypic changes, the total Chl ($\chi^2 = 10.833$, $df = 4$, $p = 0.029$), Chl a ($\chi^2 = 11.300$, $df = 4$, $p = 0.023$) and Chl b ($\chi^2 = 9.567$, $df = 4$, $p = 0.048$) content decreased by 60.5, 67.4 and 43.2% at 20 μL L$^{-1}$ $\alpha$-pinene, respectively (Figs. 2A–2C).

**Table 2  Effects of $\alpha$-pinene on height, FW, DW and RWC of drooping wildryegrass seedlings exposed to 0, 5, 10, 15 and 20 $\mu$L L$^{-1}$ of $\alpha$-pinene for a period of 4 days (with Kruskal–Wallis test).** Fresh weight (FW), dry weight (DW) and relative water content (RWC). The values are mean $\pm$ standard error ($n = 3$). Different letters indicate comparisons with significant difference ($p < 0.05$) among treatments.

| $\alpha$-pinene ($\mu$L L$^{-1}$) | Plant height (cm) | FW (g seedlings$^{-15}$) | DW (g seedlings$^{-15}$) | Leaf RWC (%) |
|---|---|---|---|---|
| 0 | $14.17 \pm 0.05$a | $0.575 \pm 0.02$a | $0.065 \pm 0.002$a | $84.08 \pm 1.62$b |
| 5 | $14.36 \pm 0.07$a | $0.657 \pm 0.04$a | $0.064 \pm 0.004$a | $87.85 \pm 3.82$ab |
| 10 | $14.09 \pm 0.06$a | $0.569 \pm 0.03$a | $0.055 \pm 0.001$ab | $102.94 \pm 7.12$ab |
| 15 | $14.13 \pm 0.05$a | $0.546 \pm 0.05$a | $0.051 \pm 0.002$b | $119.82 \pm 9.20$a |
| 20 | $13.10 \pm 0.07$a | $0.489 \pm 0.04$a | $0.046 \pm 0.003$b | $118.60 \pm 4.29$a |

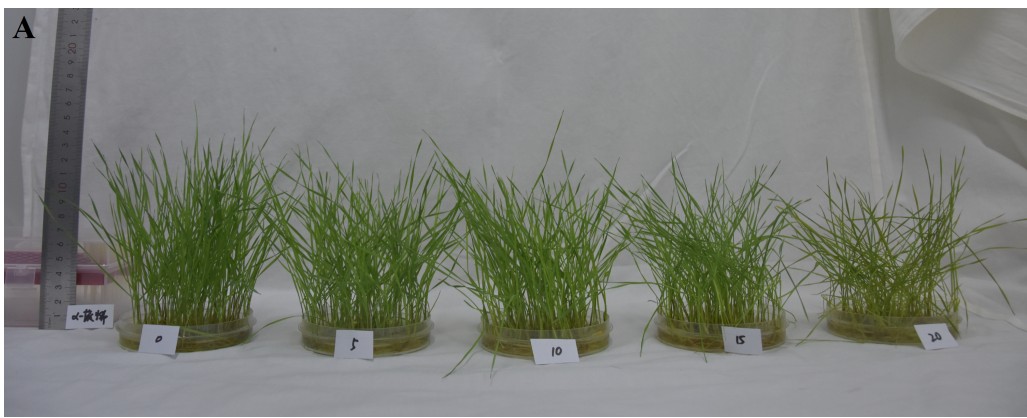

**Figure 1  Effects of $\alpha$-pinene on toxicity symptoms in the leaves of drooping wildryegrass seedlings subjected to 0, 5, 10, 15 and 20 $\mu$L L$^{-1}$ $\alpha$-pinene for 4 days.**

## Effects of $\alpha$-pinene on water-soluble proteins, soluble sugars

The effects of $\alpha$-pinene on water-soluble proteins and the soluble sugars showed similar change trend (Figs. 3A and 3B). The water-soluble proteins ($\chi^2 = 12.900$, $df = 4$, $p = 0.012$) and the soluble sugars ($\chi^2 = 13.033$, $df = 4$, $p = 0.011$) levels increased significantly at 5, 10, 15 and 20 $\mu$L L$^{-1}$ $\alpha$-pinene, respectively, but no significant differences between 10, 15 and 20 $\mu$L L$^{-1}$ $\alpha$-pinene were detected (Figs. 3A and 3B).

## Effects of $\alpha$-pinene on H$_2$O$_2$ accumulations, MDA levels and pro contents

No significant differences in H$_2$O$_2$ levels at different doses of $\alpha$-pinene treatment (Fig. 4A), but caused membrane damage. The contents of lipid peroxidation product MDA and cell membrane damage product Pro in the seedlings increased sharply when $\alpha$-pinene concentration $\geq$15 $\mu$L L$^{-1}$ (Figs. 4B and 4C). A remarkable increase of MDA level by 253.0% at 20 $\mu$L L$^{-1}$ $\alpha$-pinene ($\chi^2 = 11.567$, $df = 4$, $p = 0.021$; Fig. 4B). Pro content had a steady increase with $\alpha$-pinene concentrations ($\chi^2 = 13.500$, $df = 4$, $p = 0.009$; Fig. 4C).

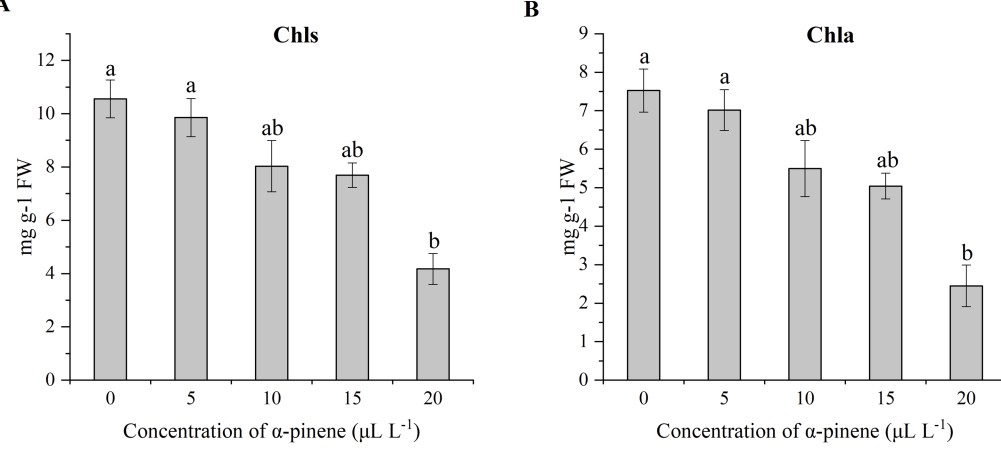

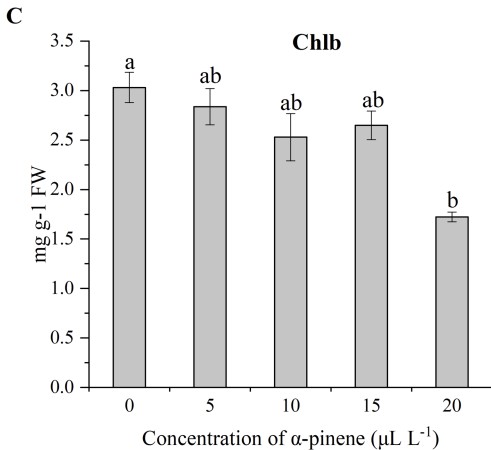

**Figure 2** Effects of $\alpha$-pinene on photosynthetic pigment in the leaves of drooping wildryegrass seedlings subjected to 0, 5, 10, 15 and 20 $\mu$L L$^{-1}$ $\alpha$-pinene for 4 days (with Kruskal–Wallis test). (A) Total chlorophylls (Chls). (B) Chlorophyll a (Chla). (C) Chlorophyll b (Chlb). Fresh weight (FW). Different letters indicate comparisons with significant difference ($p < 0.05$) among treatments. The values are means $\pm$ standard error ($n = 3$).

## Effects of $\alpha$-pinene on ROS-metabolizing enzymes

The antioxidant system, i.e., enzyme defense system of drooping wildryegrass seedlings, plays a crucial part in the oxidative stress induced by $\alpha$-pinene. SOD activity showed a unimodal variation with $\alpha$-pinene concentration and the maximum value appeared at 5 $\mu$L L$^{-1}$ $\alpha$-pinene (72.6%). No significant SOD activity differences between 0, 10, 15 and 20 $\mu$L L$^{-1}$ $\alpha$-pinene ($\chi^2 = 11.033$, $df = 4$, $p = 0.026$; Fig. 5A). CAT activity decreased following different concentrations of $\alpha$-pinene treatment ($\chi^2 = 12.367$, $df = 4$, $p = 0.015$; Fig. 5B). POD activity increased by 94.4% at 15 $\mu$L L$^{-1}$ $\alpha$-pinene ($\chi^2 = 11.067$, $df = 4$, $p = 0.026$; Fig. 5C). APX activity increased by 98.3, 161.7 and 180.7% at 5, 10 and 15 $\mu$L L$^{-1}$ $\alpha$-pinene, respectively; however, this increasing trend started to decrease, showing 53.4% increase at 20 $\mu$L L$^{-1}$ $\alpha$-pinene ($\chi^2 = 12.767$, $df = 4$, $p = 0.012$; Fig. 5D).

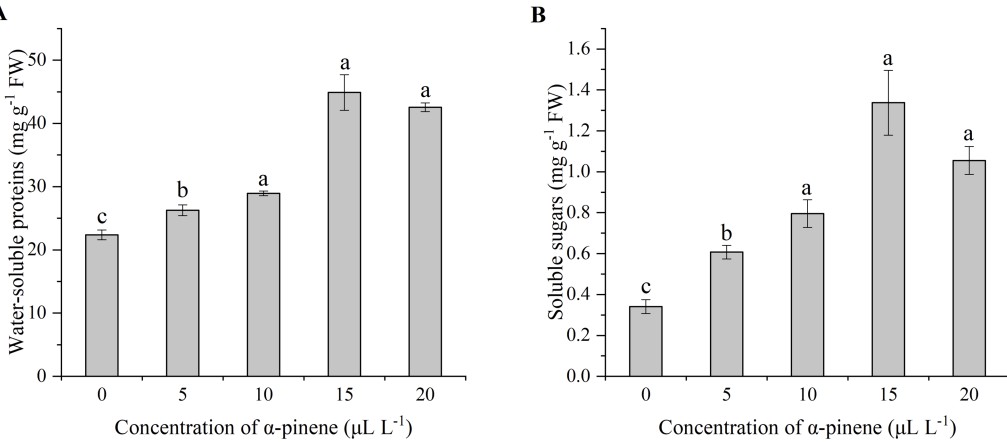

**Figure 3** Levels of water-soluble proteins and soluble sugars in the leaves of drooping wildryegrass seedlings subjected to 0, 5, 10, 15 and 20 μL L$^{-1}$ α-pinene for 4 days (with Kruskal–Wallis test). (A) Water-soluble proteins. (B) Soluble sugars. Fresh weight (FW). Different letters indicate comparisons with significant difference ($p < 0.05$) among treatments. The values are means ± standard error ($n = 3$).

### Effects of α-pinene on GSH level and GR activity
GSH level and GR activity increased at high dose of α-pinene. Compared with the control, a significant increase of GSH level at 15 and 20 μL L$^{-1}$ α-pinene ($\chi^2 = 12.000$, $df = 4$, $p = 0.017$; Fig. 6A), and GR activity at 20 μL L$^{-1}$ α-pinene ($\chi^2 = 9.800$, $df = 4$, $p = 0.044$; Fig. 6B).

### Effects of α-pinene on nitrogen metabolites
The level of NO in the drooping wildryegrass leaves increased by 308.9 and 1545.8% at 15 and 20 μL L$^{-1}$ α-pinene, as compared with untreated control ($\chi^2 = 12.533$, $df = 4$, $p = 0.014$; Fig. 7A). No significant differences for NR activity was detected at different doses of α-pinene (Fig. 7B).

### Endogenous hormone levels
The endogenous levels of ABA, Zeatin, SA, GA4, GA7, JA and IAA level in drooping wildryegrass seedling leaves following α-pinene treatment varied with concentrations. The ABA level increased significantly at 5, 10, 15 and 20 μL L$^{-1}$ α-pinene, but no significant differences between 10, 15 and 20 μL L$^{-1}$ α-pinene ($\chi^2 = 12.933$, $df = 4$, $p = 0.012$; Fig. 8A). A significant increase of Zeatin level was recorded at 20 μL L$^{-1}$ α-pinene ($\chi^2 = 11.500$, $df = 4$, $p = 0.021$; Fig. 8B). The SA ($\chi^2 = 11.433$, $df = 4$, $p = 0.022$; Fig. 8C) and JA ($\chi^2 = 12.833$, $df = 4$, $p = 0.012$; Fig. 8F) level increased by 125.8, 138.2% and 90.0, 177.9 times at 15 and 20 μL L$^{-1}$ α-pinene, respectively. No significant differences were found in GA4 levels between different α-pinene treatments (Fig. 8D). GA7 level increased by 371.5% at 20 μL L$^{-1}$ α-pinene ($\chi^2 = 13.033$, $df = 4$, $p = 0.011$; Fig. 8E). IAA levels increased by 236.9, 556.3 and 1202.9% at 10, 15 and 20 μL L$^{-1}$ α-pinene ($\chi^2 = 12.967$, $df = 4$, $p = 0.011$; Fig. 8G).

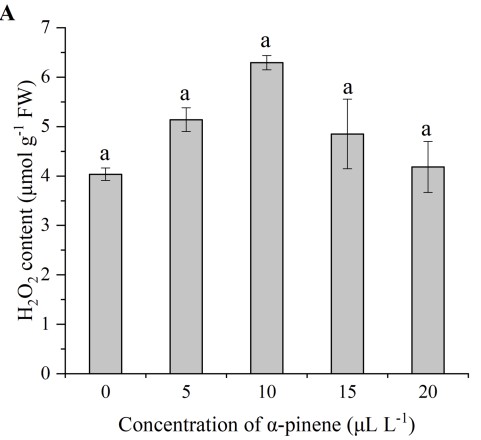

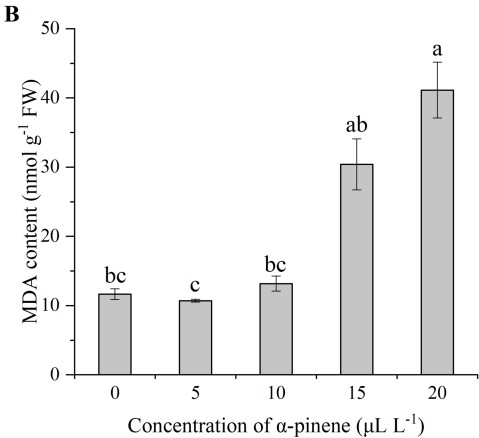

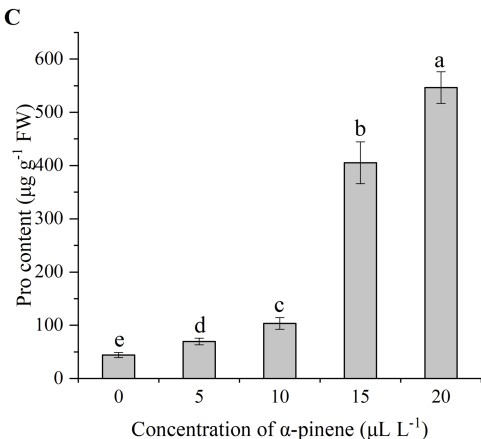

**Figure 4 Reactive oxygen species (ROS) generation and lipid peroxidation in the leaves of drooping wildryegrass seedlings subjected to 0, 5, 10, 15 and 20 μL L$^{-1}$ α-pinene for 4 days (with Kruskal–Wallis test).** (A) Hydrogen peroxide ($H_2O_2$). (B) Malondialdehyde (MDA). (C) Proline (Pro). Fresh weight (FW). Different letters indicate comparisons with significant difference ($p < 0.05$) among treatments. The values are means ± standard error ($n = 3$).

## DISCUSSION

In the long-term evolution process, plants respond to all kinds of environmental stresses through a signal regulation mechanism to maintain normal growth (*Chen & Yang, 2020*). Generally, environmental stresses have detrimental effects on plant growth, stress proteins, stress hormones, and stress metabolites synthesis. Allellochemicals, the phytotoxins released from plants, exert inhibition on growth of plants, like *Metasequoia glyptostroboides* water extracts on *Lepidium sativum*, *Lactuca sativa*, *Medicago sativa* (*Matuda et al., 2022*); *Tithonia diversifolia* water extract on neighboring plants (*Kato-Noguchi, 2020*); and *Rhus typhina* water extracts on *Tagetes erecta* (*Qu et al., 2021*). In the present report, α-pinene treated seedlings had no significant influences on plant height and FW (Table 2), but the increased applications of α-pinene inhibited the biomass of drooping wildryegrass (Table 2). Additionally, the balanced water status of plants was broken and seedling development

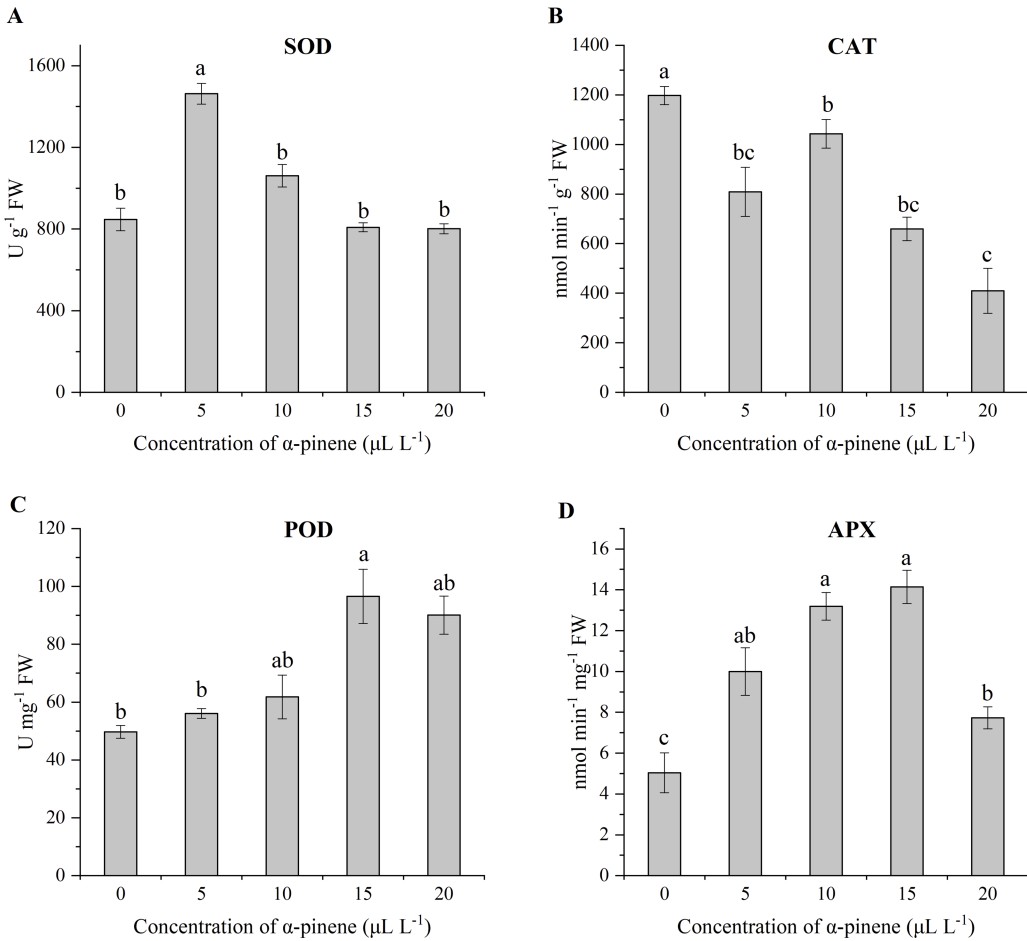

**Figure 5** **Activities of reactive oxygen species (ROS)-detoxifying enzymes in the leaves of drooping wildryegrass seedlings subjected to 0, 5, 10, 15 and 20 $\mu$L L$^{-1}$ $\alpha$-pinene for 4 days (with Kruskal–Wallis test).** (A) Superoxide dismutase (SOD). (B) Catalase (CAT). (C) Peroxidase (POD). (D) Ascorbate per-oxidase (APX). Fresh weight (FW). Different letters indicate comparisons with significant difference ($p < 0.05$) among treatments. The values are means $\pm$ standard error ($n = 3$).

was inhibited under various abiotic stresses (*Mostofa et al., 2017*). The RWC presented a significant increase at 15 and 20 $\mu$L L$^{-1}$ $\alpha$-pinene (Table 2), suggesting allelochemicals may damage cell membranes through direct or indirect interaction (*Yu et al., 2003*). We guess this phenomenon is related to the transparent closed tank, when the membrane system of drooping wildryegrass was destroyed at 15 and 20 $\mu$L L$^{-1}$ $\alpha$-pinene, the seedlings could absorb more water at high humidity atmospheres. The changes in Chls was consistent with the phenotype in various abiotic stresses (Fig. 1). The $\alpha$-pinene drastically affected Chls, Chl a and Chl b biosynthesis at 20 $\mu$L L$^{-1}$ (Figs. 2A, 2B and 2C), indicating that biomass and cell membranes of drooping wildryegrass were inhibited and destroyed at 15 and 20 $\mu$L L$^{-1}$ $\alpha$-pinene. Protein and sugar are two important macromolecules that provide metabolites and energy through various biochemical processes to strengthen plant immunity during the onset of stress (*Krasensky & Jonak, 2012*). In our study, total water-soluble proteins

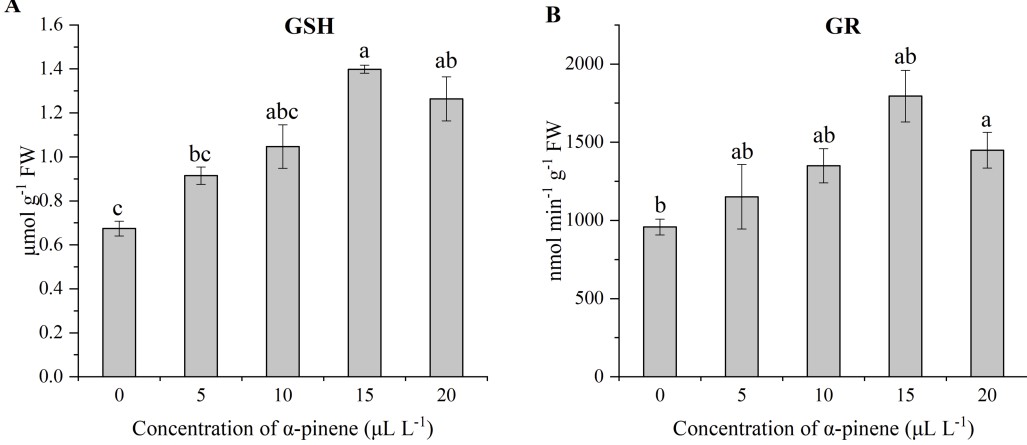

**Figure 6** **Levels of GSH and activities of GR in the leaves of drooping wildryegrass seedlings subjected to 0, 5, 10, 15 and 20 μL L⁻¹ α-pinene for 4 days (with Kruskal–Wallis test).** (A) glutathione (GSH). (B) glutathione reductase (GR). Fresh weight (FW). Different letters indicate comparisons with significant difference ($p < 0.05$) among treatments. The values are means ± standard error ($n = 3$).

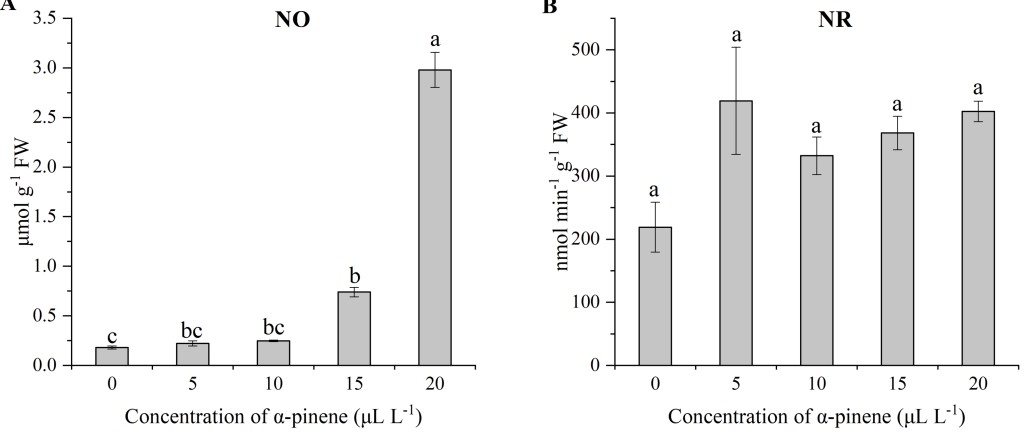

**Figure 7** **Effects of α-pinene on nitrogen metabolites in the leaves of drooping wildryegrass seedlings subjected to 0, 5, 10, 15 and 20 μL L⁻¹ α-pinene for 4 days (with Kruskal–Wallis test).** (A) nitric oxide (NO). (B) nitrate reductase (NR). Fresh weight (FW). Different letters indicate comparisons with significant difference ($p < 0.05$) among treatments. The values are means ± standard error ($n = 3$).

and soluble sugars were accumulated significantly at 5, 10, 15 and 20 μL L⁻¹ α-pinene, suggesting drooping wildryegrass rapidly synthesized various stress-responsive proteins and sugars to combat α-pinene toxic effects to some extent (Figs. 3A and 3B). Similar results were also reported in self-allelopathy of *Casuarina equisetifolia* seedlings (*Lin, 2007*).

ROS are one of the most classical signaling molecules and response to environmental stress in plants (*Chen & Yang, 2020*). ROS include several types of active molecules, such as superoxide anion radical ($O_2^{·-}$), hydrogen peroxide ($H_2O_2$), hydroxyl radical ($OH^-$) and singlet oxygen ($^1O_2$) (*Noctor, Reichheld & Foyer, 2018*). The $O_2^{·-}$ can be spontaneously

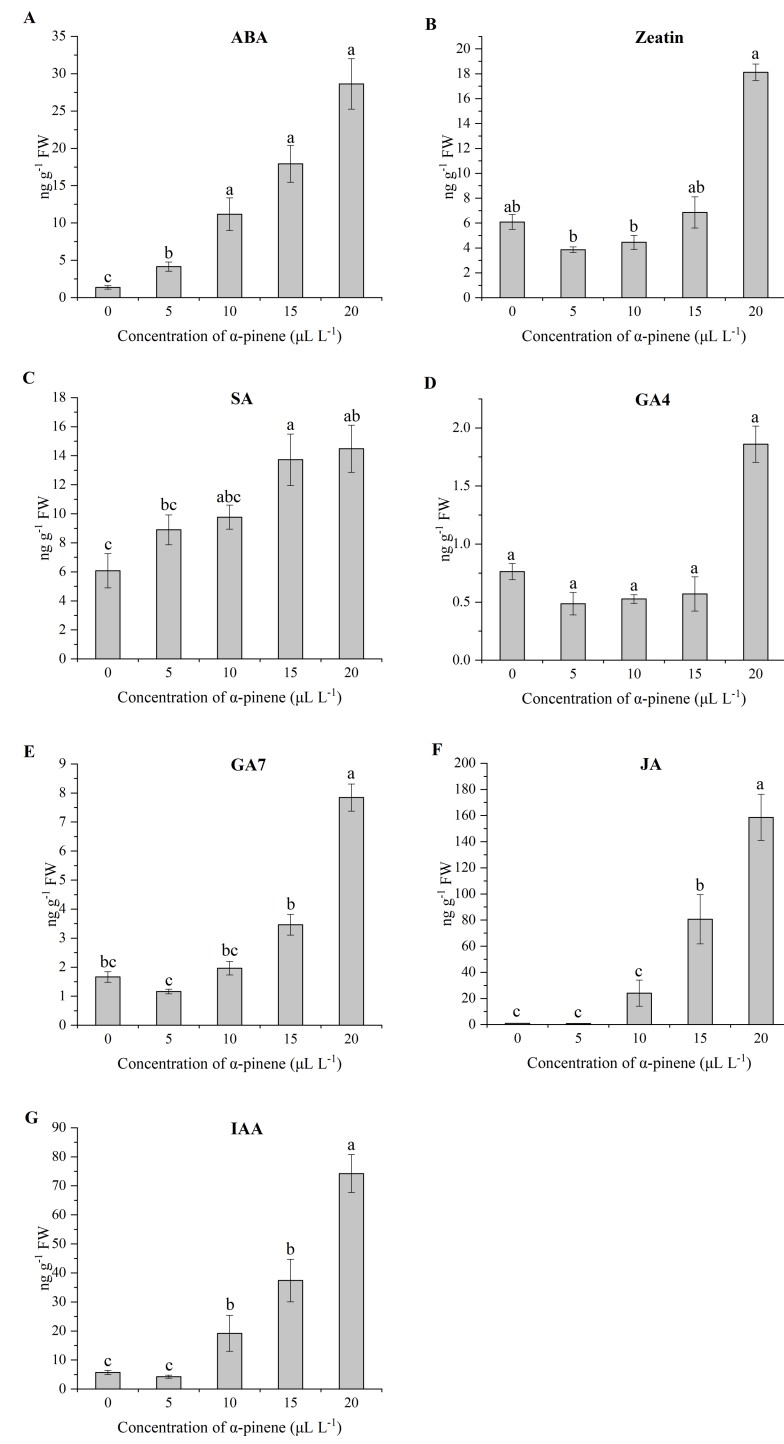

**Figure 8** **Levels of endogenous hormone in the leaves of drooping wildryegrass seedlings subjected to 0, 5, 10, 15 and 20 μL L⁻¹ α-pinene for 4 days (with Kruskal–Wallis test).** (A) Abscisic acid (ABA). (B) Zeatin. (C) Salicylic acid (SA). (D) Gibberellin 4 (GA4). (E) Gibberellin 7 (GA7). (F) Jasmonic acid (JA). (G) Indole acetic acid (IAA). Fresh weight (FW). Different letters indicate comparisons with significant difference ($p < 0.05$) among treatments. The values are means ± standard error ($n = 3$).

and rapidly inverted to $H_2O_2$, and can also be disproportionated by SOD which detoxify superoxide anion to $H_2O_2$ by enzymatic reaction (*Chen & Yang, 2020*). In addition, APX, CAT are ROS detoxifying proteins, and GSH is an antioxidant (*Mittler et al., 2004*). GSH maintains redox balance inside cells, including anti-oxidation, free radical scavenging, electrophile elimination, and may directly react with ROS (*Thiboldeaux, Lindroth & Tracy, 1998*). GR plays a crucial part in the control of the intracellular redox environment by catalyzing the reduction of oxidised glutathione (GSSG) to GSH (*Coelho et al., 2017*). GSH and GR were involved in ascorbate-glutathione (AsA-GSH) cycle, which has been recognized to be related to oxidative stress (*Foyer & Noctor, 2011*). MDA is a widely used marker of oxidative lipid injury (*Davey et al., 2005*). In our present study, we observed fast accumulation of MDA in drooping wildryegrass leaves at 20 µL L$^{-1}$ $\alpha$-pinene (Fig. 4B), indicating that high dosage of $\alpha$-pinene caused oxidative damage system of drooping wildryegrass. The other allelochemical also triggers a wave of oxidative damage (*Bais et al., 2003*). In many plants, free Pro accumulates in response to various abiotic stresses. Pro can stabilise subcellular structures and scavenge free radicals (*Hare & Cress, 1997*). Pro content had a significant increase in response to $\alpha$-pinene stress at 5 and 10 µL L$^{-1}$. However, a sharp increase in Pro content indicated that drooping wildryegrass seedlings was seriously affected at 15 and 20 µL L$^{-1}$ $\alpha$-pinene. The increased activity of the antioxidant enzymes exhibited different kinetics of seedlings growth during the dose gradient treatment of $\alpha$-pinene. The enzyme system plays an active role in inhibiting the production of $H_2O_2$ in drooping wildryegrass leaves (Fig. 4A). The changes in antioxidants suggested that drooping wildryegrass seedlings were sensitive to $\alpha$-pinene, as SOD and APX increased at 5 µL L$^{-1}$ $\alpha$-pinene to resist stress (Figs. 5A and 5D). The activity of APX increased with $\alpha$-pinene dose increased, indicating that the plant produced APX decreased the membrane lipid peroxidation quickly at 10 µL L$^{-1}$ $\alpha$-pinene (Fig. 5D). POD participate in the removal of $H_2O_2$ from plant cells (*De Gara, 2004*). The high-activity of POD, APX along with the high level of GSH found at 15 µL L$^{-1}$ $\alpha$-pinene indicated that the AsA-GSH cycle may contribute to the cellular redox equilibrium (Figs. 5C, 5D and 6A). However, when growth of seedlings was severely stressed at 20 µL L$^{-1}$ $\alpha$-pinene, the activity of POD, GR and level of GSH remained stable, the activity of APX started declining, growth of seedlings was inhibited (Figs. 5D, 6A and 6B). Contrary to the other antioxidant enzymes and antioxidants, the activity of CAT decreased at different doses of $\alpha$-pinene (Fig. 5B). Therefore, when drooping wildryegrass seedlings is stressed by $\alpha$-pinene, SOD and APX played the pioneer role in the low concentration. With the increase of $\alpha$-pinene concentration, APX, POD and GSH played a bigger active role. When the stress degree was maximum, POD, GR activity and GSH level remained stable. The dynamic changes of the enzyme system cleared $H_2O_2$ produced under $\alpha$-pinene stress conditions. The change of detoxifying enzyme system may be the mechanisms that allelopathy, as reported in *Oryza sativa* (*Fang et al., 2008*) and *Citrullus lanatus* (*Geng et al., 2005*).

NO is an endogenous signal that responses to several stimuli in plants (*Neill et al., 2008*; *He et al., 2022*). NO was associated with the responses to abiotic stress in plants, such as drought and heat stress (*Leshem, Wills & Ku, 1998*). The increase of NO level has also been found in allelopathic effects of some weed species (*Xie et al., 2021*). NO also enhances the

activity of the enzyme through some unidentified signaling pathways. NO may increase the antioxidant capacity of cells by increasing the activities of APX (*Neill et al., 2008*). In our study, NO level increased significantly from 15 $\mu$L L$^{-1}$ $\alpha$-pinene (Fig. 7A). The increase of APX activity may be related to the increase of NO level at 15 $\mu$L L$^{-1}$ $\alpha$-pinene. NO is catalysed by nitrate reductase (NR) under certain conditions (*Kaiser & Huber, 2001*). However, $\alpha$-pinene treatment had no effect on NR activity (Fig. 7B). The increase of NO level was not related to NR. ABA triggers NO generation (N et al., 2008). We guess that the increase of NO level may be related to the increase of ABA levels (Figs. 7A and 8A).

Plants have evolved a variety of stress responses, and the changes of plant hormone were different when plants respond to different stress condition (*Verma, Ravindran & Kumar, 2016*). However, hormones are related by synergistic or antagonistic cross-talk and they regulate each other's biosynthesis process (*Peleg & Blumwald, 2011*). The hormone levels we studied were altered by $\alpha$-pinene stress. Typically, ABA is closely associated with abiotic stress defense plants, and ABA levels increased under drought, salinity, cold, heat stress and wounding conditions (*Lata & Prasad, 2011*; *Zhang et al., 2006*). It was reported that the allelochemicals stimulation increased ABA levels (*Bogatek & Gniazdowska, 2007*). In our study, ABA level showed a significant increase at different $\alpha$-pinene doses (Fig. 8A). Phenolic allelochemicals ferulic acid also activated the synthesis of ABA (*Holappa & Blum, 1991*). Research in *Arabidopsis thaliana* revealed that numerous genes encoding proteins associated with cytokinins (CKs) signaling pathways that were differentially affected by various abiotic stresses (*Argueso, Ferreira & Kieber, 2010*). CKs levels in plants may increase or decrease under water limiting conditions (*Argueso, Ferreira & Kieber, 2010*). Zeatin and its derivatives are the most important group of isoprenoid CKs (*Gajdošová et al., 2011*). In this study, the levels of Zeatin decreased at 5 $\mu$L L$^{-1}$ $\alpha$-pinene and increased at 20 $\mu$L L$^{-1}$ $\alpha$-pinene. There were no significant differences in Zeatin levels compared with the control treatment. However, there was a significant difference in Zeatin levels at 5 $\mu$L L$^{-1}$ $\alpha$-pinene and 20 $\mu$L L$^{-1}$ $\alpha$-pinene, indicating that there was a difference between the synthesis mechanisms at low and high concentrations of $\alpha$-pinene (Fig. 8B). The increased level of CKs could inhibit leaf senescence during stress conditions and might increase the level of Pro (*Alvarez et al., 2008*). The increase in Zeatin level may be attributed to an increase in Pro level at 20 $\mu$L L$^{-1}$ $\alpha$-pinene (Fig. 4C). CKs can rapidly induce NO biosynthesis in plant cell cultures of Arabidopsis, parsley and tobacco (*Tun, Holk & Scherer, 2001*). We guess that the increased NO level was also related to the accumulation of Zeatin at 20 $\mu$L L$^{-1}$ $\alpha$-pinene (Figs. 7A and 8B). SA is a signal molecule involved in plant defense responses (*Shah, 2003*). In our study, SA level showed a significant increase at 15 and 20 $\mu$L L$^{-1}$ $\alpha$-pinene (Fig. 8C), as supported by the studies on abiotic stress, like drought (*Pandey, 2017*; *Sergi & Josep, 2003*), cold (*Kosová et al., 2012*), heat (*Dat et al., 1998*) andsalinity stress (*Sawada, Shim & Usui, 2006*). Reduction of GA levels and signaling result in plant growth restriction under several stresses conditions, including cold, salt and osmotic stress (*Colebrook et al., 2014*). GA is composed of a large group of tetracyclic diterpenoid carboxylic acids, of which GA1, GA3, GA4 and GA7 mostly active (*Sponsel, 2003*). The $\alpha$-pinene treatment decreased GA1 and GA3 levels so that their levels did not reach the detection limits of the instruments. GA4 levels had no significant difference at different $\alpha$-pinene doses, and GA7 levels showed

a significant increase at high dosage of $\alpha$-pinene (20 µL L$^{-1}$) (Figs. 8D and 8E). JA play crucial roles in plant responses to abiotic stress factors, and there is growing evidence that auxin is involved in the trade-off between growth and defense. Some studies also revealed that JA increases auxin production (*Pérez-Alonso et al., 2021*). The $\alpha$-pinene treatment caused JA and IAA level to show a similar pattern of response (Figs. 8F and 8G). The result of phytohormone indicated that ABA and IAA of drooping wildryegrass seedlings leaves were sensitive to $\alpha$-pinene. Zeatin, SA, GA7 and JA levels of drooping wildryegrass seedlings could not be affected at 5 and 10 µL L$^{-1}$ $\alpha$-pinene. As the degree of $\alpha$-pinene stress, ABA and IAA levels continued to increase. SA and JA played an important role in resisting allelopathic effects at 15 µL L$^{-1}$ $\alpha$-pinene. At high dosage of $\alpha$-pinene, ABA, Zeatin, SA, GA7, JA and IAA levels increased. The $\alpha$-pinene treatment had the greatest impact on ABA and IAA levels. They act as key regulators under individual drought and pathogen stress respectively (*Gupta et al., 2017*). The mechanism of drooping wildryegrass seedlings hormone change needs further study.

## CONCLUSIONS

The $\alpha$-pinene-induced allelopathy activated physiological response of drooping wildryegrass that led to change of biomass, RWC, photosynthetic pigment, water-soluble proteins, soluble sugars, MDA, GSH levels, Pro contents, ROS-metabolizing enzymes, nitrogen metabolites and endogenous hormone levels. The $\alpha$-pinene-stress had no significant effect on height, FW, $H_2O_2$, NR and GA4. The dynamic changes of enzyme system cleared $H_2O_2$ produced under $\alpha$-pinene stress conditions. However, higher doses of $\alpha$-pinene caused severe phytotoxic effects by impairing several physiological, biochemical and phytohormone processes in drooping wildryegrass. Endogenous signal NO and cell membrane damage product Pro accumulated in leaves of drooping wildryegrass seedlings at 15 µL L$^{-1}$ $\alpha$-pinene, and lipid peroxidation product MDA accumulated at 20 µL L$^{-1}$ $\alpha$-pinene. The $\alpha$-pinene caused stress damage to drooping wildryegrass seedlings mainly through break the balance of membrane system rather than ROS production at 15 and 20 µL L$^{-1}$concentrations. Additionally, the $\alpha$-pinene treatment has the most impact on ABA and IAA levels. Drooping wildryegrass seedlings can effective in counteracting the harmful effects of ROS generated at lower doses of $\alpha$-pinene, and they were severely stressed at higher doses of $\alpha$-pinene. Our findings provided references for understanding the allelopathic mechanism of allelochemicals in plants.

## ACKNOWLEDGEMENTS

The authors would like to thank HengxiaYin for her help and anonymous reviewers for their helpful comments.

### Funding

This work was supported by the Youth Foundation of Qinghai University (No. 2020-QNY-2), the National Natural Science Foundation of China (No. 31760691) and the Programme of Introducing Talents of Discipline to Universities (No. D18013). The funders had no role in study design, data collection and analysis, decision to publish, or preparation of the manuscript.

### Competing Interests

The authors declare there are no competing interests.

### Author Contributions

- Mengci Chen conceived and designed the experiments, performed the experiments, analyzed the data, prepared figures and/or tables, authored or reviewed drafts of the article, and approved the final draft.
- Youming Qiao conceived and designed the experiments, analyzed the data, authored or reviewed drafts of the article, and approved the final draft.
- Xiaolong Quan performed the experiments, prepared figures and/or tables, and approved the final draft.
- Huilan Shi analyzed the data, authored or reviewed drafts of the article, and approved the final draft.
- Zhonghua Duan analyzed the data, authored or reviewed drafts of the article, and approved the final draft.

### Data Availability

The raw data is available in the Supplemental Files.

### Supplemental Information

Supplemental information for this article can be found online at http://dx.doi.org/10.7717/peerj.14100#supplemental-information.

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
