# Peer review of "Physiological, biochemical and phytohormone responses of Elymus nutans to α-pinene-induced allelopathy"

_PeerJ, doi:10.7717/peerj.14100_

## Round 0.1 · original submission · Major Revisions

Dear Authors

Kindly amend your manuscript as suggested by the referees.

Reviewer 1 ·

Basic reporting

The authors examined the effects of different doses α-pinene on the Physiological, biochemical and phytohormone responses changes of Elymus nutans. In general, this is a comprehensive study, the experiments were well conducted, and the presented results very convincing. However, there are yet many problems need to be addressed. In my opinion, the manuscript needs a major revision for reconsideration for possible publication in Peer J.
The english is poor, and there are many grammatical mistake in the text, I think it needs great improve.
Abreviations: the full name of the abreviations should be list for the first time it used in the manuscript,such as RWC, MDA...including that in the abstract, please check this throughout the text.
A mechanism discussion and conclusion were needed both in the abstract and the discussion sections.

Experimental design

In order to keep consistent with the treated group, did the author treated the control Elymus nutans with the same reagent that just without α-pinene? Meawhile, the method of α-pinene application should be specified in the text.

Validity of the findings

In general, this is a comprehensive study, which provides a comprehensive insight of the the α-pinene on growth of Elymus nutans.

Additional comments

Include a brief summary in some methodologies and the references related such as the assay of MDA, H2O2 and the assay of enzymes..., now it is hard to follow by the readers and other researchers.
What is the role of figure 1 in the manuscript?it is not mentioned throughout the text.
The results should be rephrased, there are many valuable datas obtained from the experiment, but it is not well presented.
Discussion should be further improved by focusing on the probable mechanism of α-pinene in affecting the overall growth of Elymus nutans.
Specific comments:
Line 47: what is the mean of SA and ABA level increased? in different samples or as the dose of α-pinene application increased?
Line123: replace “treatment” by “treated”.
Line 126-127:rephrase this sentence.
Line156: delete the word ”by”.
Line 196: replace the word “a powder” by“ powder”.
Line 212: the word “p” should be italic.
Line 352:replace the word “drought stress” by“ drought”.
Line 381:delete the word “or” in the sentence.
Line 384:what is the mean of “we think wether the increased NO level......”.

Reviewer 2 ·

Basic reporting

The work is interesting in its approach of grass physiology. However it has some statistical things that should be improved. Firstly, it is not written if the assumptions of normality and homoscedascity were checked. Secondly, an n=3 seems to be an actually low number of replicates. In addition, from a graphical point of view, the figures seem not to be homoscedastic to apply an ANOVA.

Experimental design

As I have written previously, the number of replicates is low and the assumptions for an ANOVA seem not to be checked. Also, p-values are not written in the text while the results are presented.

Validity of the findings

A higher number of replicates should be used and the assumptions must be checked. In spite of this, the experiments are correctly focused from a biological perspective, buy without a proper statistical design they lose validity.

Additional comments

The authors should improve the grammar of this work. Also, discussion must be reorganized in order to present previously reported information in the introduction and analyzed the results according to previous literature in the discussion.

Annotated reviews are not available for download in order to protect the identity of reviewers who chose to remain anonymous.

Reviewer 3 ·

Basic reporting

Dear authors
I carefully reviewed the manuscript entitled “Physiological, biochemical and phytohormone responses changes of Elymus nutans resistance pinene-induced allelopathy” which is very interesting. In according with results, high concentrations of alpha pinene affected the development of Elymus nutants plants, especially the dry weight and several differences among biochemical parameters explored were observed under high doses of alpha pinene and low doses of alpha pinene added to Elymus nutans. I have several concerns about the work, first the concentrations of alpha pinene used are not clear, authors mentioned 5, 10, 15 and 20 µL/L, but what does mean those unities? I suggest to explain the form to obtain the initial extract (material and methods). In the abstract and along the paper authors refer that in low concentration of alpha pinene “The alpha pinene promoted seedling height and fresh weight of Elymus nutans”, however, data of table 2 do not shown any statistical difference in this respect. In my opinion, all the paper must be focused only about results with statistic differences and not on tendences without statistic differences. Therefore, I recommend to restructure the manuscript highlighting only results sustained with statistic differences (results and discussion).
Minor suggestions
1) Sometimes Zeatin was written using capital letter for Z and other times lowercase letter. The form of redaction must be unified.
2) Line 104. The word “biochemecal” must be redacted properly.
3) Line 134. The meaning of RWD must be defined.
4) Line 139. The meaning of BCA must be defined.
5) Line 146. A proper space must be inserted after the word method. The same for line 150 (after the word acetone) and line 152 (after the word measured).
6) Lines 149, 158. Authors wrote “÷W”. A proper space must be inserted between the symbol and the W letter.
7) Lines 177, 182, 186. Authors wrote “1nmol”. A proper space must be inserted between the numerical value and unities.
8) Line 178. A proper space must be inserted after the parenthesis “(POD-1-Y)”.
9) Line 206. A proper space must be inserted after the word “or” for “or[M-H]-)”.
10) Line 224. I suggest to eliminate the word “obviously”.

Experimental design

Experimental design is good. However, the concentrations of alpha pinene used in this work are not clear, authors mentioned 5, 10, 15 and 20 µL/L, but what does mean those unities? I suggest to explain the form to obtain the initial extract (material and methods)

Validity of the findings

I recommend to restructure the manuscript highlighting only results sustained with statistic differences (results and discussion). Several times authors make mention about tendencies but results are not statistically supported.

---

## Round 0.2 · accepted · Accept

Dear Sir

Thank you for submitting revised version.I am pleased to inform you that it has been accepted.